## METHOD

# High-throughput deep learning variant effect prediction with Sequence UNET

Alistair S. Dunham[1,2*] , Pedro Beltrao[1,3] and Mohammed AlQuraishi[4*]

*Correspondence:
ad44@sanger.ac.uk;
ma4129@cumc.columbia.edu

[1] European Molecular
Biology Laboratory, European
Bioinformatics Institute (EMBL-
EBI), Wellcome Genome Campus,
Hinxton, Cambridgeshire CB10
1SD, UK
[2] Wellcome Sanger Institute,
Wellcome Genome Campus,
Hinxton, Cambridgeshire CB10
1RQ, UK
[3] Department of Biology,
Institute of Molecular Systems
Biology, ETH Zurich, 8093 Zurich,
Switzerland
[4] Department of Systems Biology,
Columbia University, New York,
NY, USA

## Abstract

Understanding coding mutations is important for many applications in biology and medicine but the vast mutation space makes comprehensive experimental characterisation impossible. Current predictors are often computationally intensive and difficult to scale, including recent deep learning models. We introduce Sequence UNET, a highly scalable deep learning architecture that classifies and predicts variant frequency from sequence alone using multi-scale representations from a fully convolutional compression/expansion architecture. It achieves comparable pathogenicity prediction to recent methods. We demonstrate scalability by analysing 8.3B variants in 904,134 proteins detected through large-scale proteomics. Sequence UNET runs on modest hardware with a simple Python package.

**Keywords:** Variant effect prediction, Deep learning, Mutation, PSSM, Pathogenicity, Machine learning

## Background

Proteins are integral to biology, driving all the molecular and cellular processes that create life as we know it. The key to their success is the ability to create complex properties from limited amino acid types, which allows the many and varied processes required in living organisms to be heritably encoded. Thus, understanding the impact of genotypic changes on proteins and the phenotypes they create is a major question in biology and medicine. The number of potential coding mutations in even a single protein meant it was impossible to measure all their consequences until recent multiplexed deep mutational scanning (DMS) assays [1], and the number of genes and species means it is still impractical to measure consequences for all of them. This makes it very important to be able to predict variant effects, both to prioritise variants for experiments and for direct use in analyses.

Most traditional variant effect predictors (VEPs) are based on sequence conservation, known features such as binding sites and structural models. Many prediction tools take advantage of the natural experiment performed by evolution, using multiple sequence

alignments (MSAs) to measure positional variation across species or individuals and estimate variant effects. For instance, SIFT4G [2], EVCouplings [3] and MutationAssessor [4] are based entirely on conservation. Structure is also thought to be an important feature because structure determines protein function. Structure based models include FoldX [5] and RoseTTA [6], which use force field models to estimate variants' impact on structural stability. These have previously been limited to proteins with high quality experimental or homology models, but recent developments in structure prediction, particularly AlphaFold2 [7], makes them more widely applicable [8]. Conservation and structure can also be combined by machine learning models, alongside other knowledge such as active sites, post-translational modifications and binding motifs. Machine learning predictors include PolyPhen2 [9], Envision [10] and Condel [11]. VEPs like these have been very impactful, allowing larger analyses and prioritising and interpreting experiments.

Neural networks have been successfully applied to protein sequence tasks, including VEP. The lack of large scale labelled pathogenic variants makes directly training a deep learning VEP difficult, meaning they generally also use evolutionary conservation as a proxy for deleteriousness. For example, DeepSequence [12] learns a deep generative model for a sequence family that aims to capture the probability of observing each mutant sequence and whose posteriors can therefore be used to predict pathogenicity. It was consistently found among the top VEPs across a diverse set of DMS results [12, 13]. It has been refined in the Bayesian Variational Autoencoder (VAE) EVE model [14], which outperformed previous methods on human DMS data and ClinVar variants. However, these models still require MSA based training for each protein of interest, making them very computationally demanding. Protein language models, for example UniRep [15], AminoBert [16] or ESM-1b [17], also use unsupervised learning to capture position specific representations, which relate to a range of properties, including conservation, structural stability and secondary structure. They are trained to predict the identity of masked amino acids across many different sequences, meaning they learn general protein sequence properties. These produce sequence representations in a single forward pass but are still computationally intensive and often require top models to be trained for downstream applications, which is time consuming due to the large model size.

An intermediate approach between capturing variation in a single protein family, as EVE does, and a general protein language model is to predict per position variant frequencies for any sequence, using labelled MSA training data. This frequency defines the position's position specific scoring matrix (PSSM), summarising the cross-species diversity and conservation of the sequence. This approach balances capturing additional information contained in specific MSAs with general applicability, being able to predict rapidly from any input sequence. The link with conservation means such predictions could be used to predict deleteriousness directly or the model can be further fine-tuned using smaller scale labelled pathogenicity data. We apply this approach, presenting a fast, scalable deep learning predictor, Sequence UNET, and a corresponding python package. It uses a fully convolutional architecture to predict protein PSSMs from wild-type sequence with optional structural input. The model is trained to directly predict variant frequency or to classify low frequency variants, as a proxy for deleteriousness, and then fine-tuned for pathogenicity prediction. It outperforms previous de-novo

PSSM predictors, such as SPBuild, accurately classifies low frequency variants and achieves high VEP performance but with greater scalability. Further, our model has comparable performance on these tasks to models based on the much larger ESM-1b protein language model. These language models are much slower and require significantly more compute power as well as additional top model training time. We demonstrate the benefits of performance and scalability by rapidly calculating all possible variants for 904,134 proteins detected in a pan-genome proteomics analysis [18], something that would be impossible or prohibitively time consuming with previous VEPs.

## Results

### Sequence UNET model architecture

We have developed a highly scalable VEP, Sequence UNET (Fig. 1A), that uses a fully convolutional neural network (CNN) architecture to achieve computational efficiency and independence from length. Convolutional kernels also naturally integrate information from nearby amino acids. Since long range interactions frequently generate protein properties, we also designed the model to integrate distant information using a U-shaped compression/expansion architecture inspired by the U-NET image segmentation network [19]. Max pooling creates successively smaller layers that draw information from wide regions and the final classification is built up by processing features from each depth in turn, integrating information from a wide receptive field. Since protein structure contains information that is extremely difficult to extract from sequence alone the network supports an optional graph convolutional neural network [20] (GraphCNN) module to summarise positional structural features, which are then concatenated with

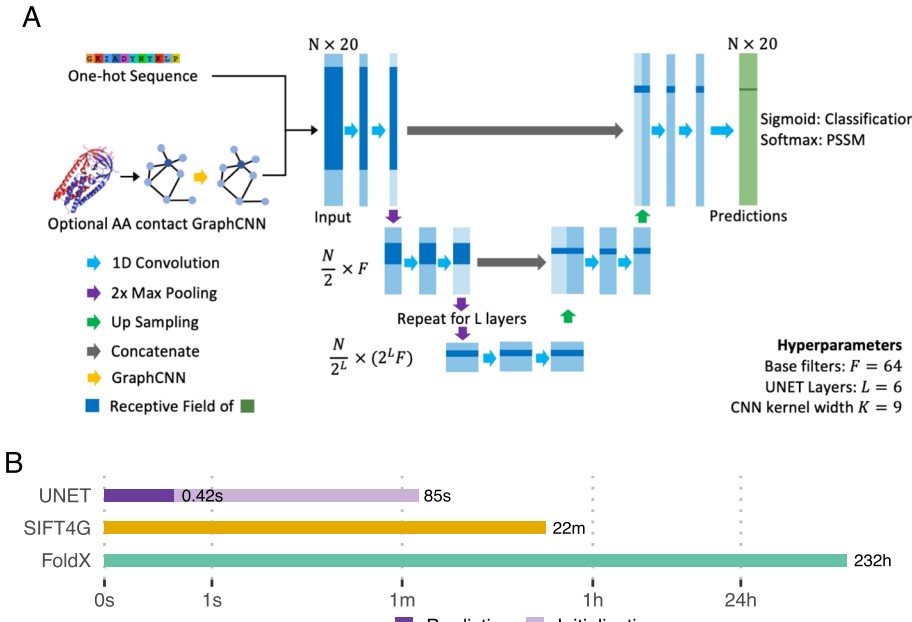

**Fig. 1** Model overview. **A** Sequence UNET model schematic. Blue rectangles represent intermediate layer output matrices and green the final prediction. **B** Bar chart showing the computation time taken to compute predictions for all variants in SARS-CoV-2 Spike protein by two commonly used tools, SIFT4G and FoldX, and Sequence UNET. These tools were chosen for a previous analysis, but broadly span the typical timescales of current tools

the one hot encoding of the wild-type sequence. We provide a more detailed description of the model and its inputs and outputs in the "Methods" section.

The model outputs a matrix of per position features and can therefore be trained to predict various positional properties. We demonstrate two related VEP use cases: predicting a matrix of probabilities that each possible variant is rare, as a proxy for deleteriousness; or directly predicting their frequency based on the input wild-type sequence. We trained using ProteinNet [21], which is a large collection of protein sequence and structure information, containing data from 104,059 PDB structures from a wide range of protein families and species alongside matching variant frequencies from large MSAs based on 332,283,871 sequences. It is designed for machine learning applications and includes in-built training/validation/testing data splits based on sequence similarity and the CASP competition [22]. However, the focus on proteins with structural information may also create a bias that reduces performance on protein types that are difficult to characterise structurally. The test set is drawn from CASP12 target proteins, which have few if any related sequences included in the training data, meaning it creates a challenging test of the models ability to generalise to unseen and often unusual sequence space. Training is very consistent, with a variance less than $10^{-5}$ in both validation loss and accuracy over 10 replicates.

The trained models are highly efficient, allowing faster and larger scale prediction than comparable tools (Fig. 1B). For example, SIFT4G took 22 min and FoldX 232 h to predict scores for all 24,187 possible variants of SARS-CoV-2 Spike protein [23]. The majority of VEPs fall within this range, with most requiring a computation intensive step such as structural sampling or MSA generation. The two most accurate neural network VEPs, EVE and DeepSequence, require both an MSA and training their latent variable model for each protein. In contrast, Sequence UNET took 420 ms to compute predictions after an 85 s initialisation time (only required once per session). This enables larger scale analyses on compute clusters and rapid analyses on desktop hardware, saving valuable researcher time and resources.

### PSSM prediction and frequency classification

We trained two base Sequence UNET models, optimising performance for PSSM prediction using a softmax output layer and Kullbeck-Leibler divergence loss and variant frequency classification using a sigmoid output and binary cross entropy. Hyperparameters were tuned in both modes with the same results (selected parameters in Fig. 1A, Additional file 1: Fig. S1). The PSSMs predicted by the model closely resemble true results (Additional file 1: Fig. S2A-B) and the frequency classifier significantly separates rare and common variants (Additional file 1: Fig. S3A-B). We find our models PSSM results correlate more strongly to true values in the ProteinNet test set than SPBuild [24] (a state of the art de novo LSTM PSSM predictor), the amino acid propensities predicted by ESM-1b and the results from a top model using ESM-1b representations trained on ProteinNet CASP12 95% thinned data (Fig. 2A, Additional file 1: Fig. S2C-D). Interestingly ESM-1b logits correlate much better with raw frequencies than normalised PSSMs, potentially because they are trained to identify the most likely amino acid at a position, not differentiate between the lower frequencies that are important for PSSMs. Including structural features slightly increases performance ($\rho = 0.472$ vs $\rho = 0.451$). Similar

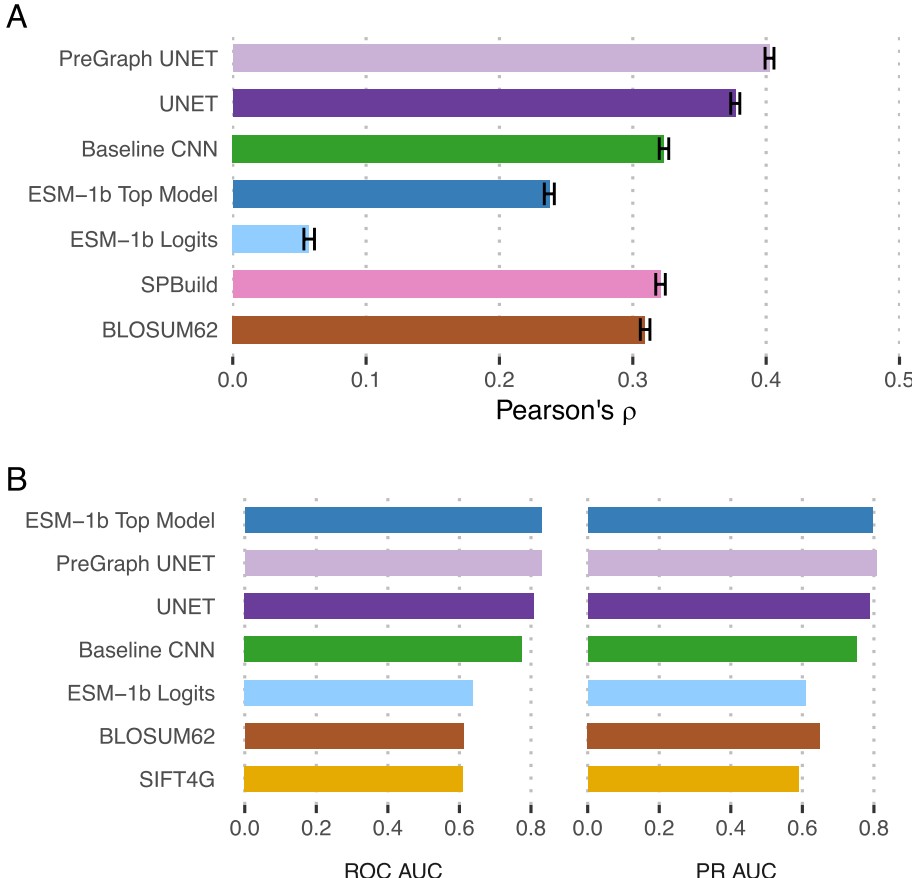

**Fig. 2** Base model prediction performance. **A** Pearson correlation between predicted and true PSSM values comparison PSSM prediction performance for Sequence UNET, a single layer CNN, SPBuild, ESM-1b logits, an ESM-1b top model and BLOSUM62. **B** ROC and PR curve AUC values comparing frequency classification performance of Sequence UNET with and without structural features, ESM-1b logits, an ESM-1b top model, a baseline single layer CNN, SIFT4G and BLOSUM62. All comparisons were made over the ProteinNet CASP12 test set, which contains 40 proteins with 12,094 total amino acids, giving 241,880 variant predictions with each tool

results are found for frequency classification ($f < 0.01$) over the ProteinNet CASP12 test set, with Sequence UNET achieving top performance equalling a top model using ESM-1b representations (Fig. 2B). The performance of a single layer baseline CNN model compared to Sequence UNET and ESM-1b on both tasks shows that it is relatively easy to get moderate performance on this task but much harder to push higher. We only compared to one VEP (SIFT4G) as deleteriousness is related to but not equivalent to frequency classification, instead comparing more widely on other datasets. Different frequency thresholds lead to different classification performance (Additional file 1: Fig. S3D), suggesting very rare or common variants are easy to classify but intermediates are more challenging. We use the $f < 0.01$ classifier and the PSSM predictor as the base for further comparisons and generalisation. $f < 0.01$ was the most challenging threshold and a common cutoff for deleteriousness, so it provides a lower bound for performance in a useful context. Further details on hyperparameter tuning and base model PSSM prediction and frequency classification performance are available in the Additional File 1: Supplementary Information.

### Generalising sequence UNET

Having shown good PSSM prediction and classification performance, we next sought to show Sequence UNET generalises to predicting deleterious variants and compare performance to other tools. We tested generalisation on three datasets: labelled human protein variants from ClinVar, standardised deep mutational scanning (DMS) data [25] and a set of gold standard *S. cerevisiae* variant classifications [26].

The model can also be fine-tuned to new tasks with additional training on external data, either refining the existing weights (fine-tuning) or replacing the final classification layer with a freshly initialised one (a top model). This creates a new model specialised to the new task while taking advantage of the general sequence features learnt in the original model and is particularly useful to transfer learning from a large general dataset to a smaller more specific one. We trained top models and fine-tuned models classifying the pathogenicity of all possible variants based on a wild-type sequence. Models were trained with and without structural features on top of the Sequence UNET frequency classification model. ClinVar was used as training data, with a random 95%/0.5%/4.5% training/validation/testing split across all pathogenic and neutral variants that occur in proteins in ProteinNet CASP12 training data. The weights of all but the top two model layers were frozen to prevent overfitting. We also trained simple single layer CNN models to predict ClinVar pathogenicity and frequency classification, to provide a lower bound for machine learning solutions to this problem. The fine-tuned models specifically predict pathogenicity probabilities for each variant at all positions (Fig. 3A). These predictions tend to be more similar for variants at a position, including the wild-type amino acid than the results of the frequency classification and PSSM models. This is partially because pathogenicity is related to the position's properties and importance but also suggests there might not be sufficient training data available to differentiate between different variants at one position beyond the average properties of that position.

UNET top models and fine-tuned models achieve comparable performance profiles to many state of the art predictors (Fig. 3B). Performance is significantly better when only proteins with structural information are considered, suggesting this bias from the ProteinNet training data has impacted what the model learnt. A training set that was not restricted to proteins from the PDB would likely help rectify this bias. The UNET top model performs slightly better than the fine-tuning approach on variants with structure but worse on those without it, independently of whether the network utilised that information. This is perhaps because the freshly initialised final weights allow the network to learn new relationships specific to the pathogenicity of structured human proteins in ClinVar whereas fine-tuning starts from weights which capture more general relationships and refines these rather than reaching highly specific minima. The fine-tuned UNET model only performs slightly worse than the much larger ESM-1v language model [27] across all variants and performs better on variants with structural data, despite being a much smaller and more manageable network. The models that performed better than ESM-1v are all directly trained on human pathogenic variation and many are ensemble models, meaning some of their features are already trained on human variation. Consequently, their performance here is potentially inflated compared to truly novel data. Interestingly, actually utilising structural data slightly reduces the performance of the fine-tuned models, suggesting

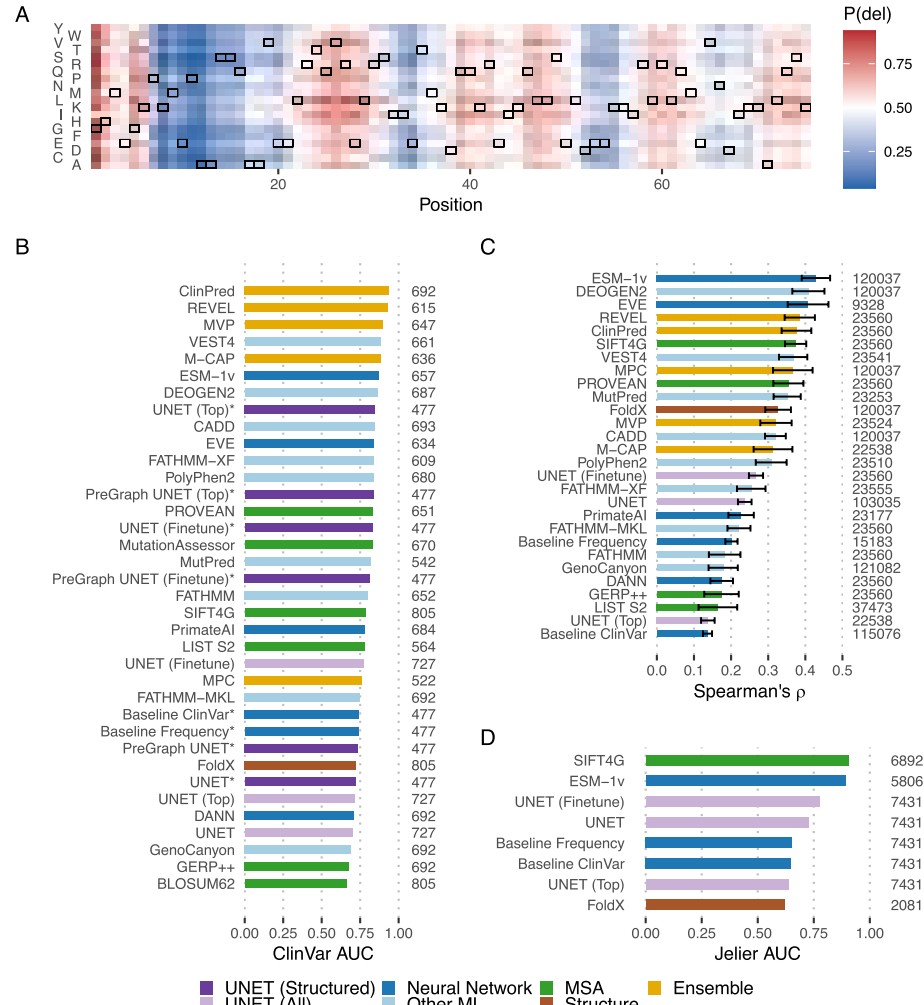

**Fig. 3** Generalising Sequence UNET. **A** Sequence UNET top model ClinVar pathogenicity predictions for the ProteinNet Casp12 test set record TBM#T0865. The wild-type amino acid at each position is outlined. **B** ROC AUC values comparing VEP performance over the ClinVar test set. Instances tested on the subset of ClinVar with structural data are marked with asterisks (*). **C** Mean and standard error of Spearman's rank correlation coefficient between VEP predictions and standardised DMS data [25]. Sequence UNET, ESM-1v, SIFT4G and FoldX predictions were available across all proteins while other tools were only available for human proteins. **D** ROC AUC values comparing performance of VEPs at classifying known deleterious and neutral *S. cerevisiae* variants [26]. In **B**, **C** and **D**, the number of variants analysed by each tool is listed on the right

it is less related to pathogenicity. The base Sequence UNET frequency classification models generalise less well to this task, although still with comparable performance to models such as FoldX. This suggests the frequency-based model does not fully capture deleteriousness without fine-tuning.

The frequency classification model was initially trained with a range of frequency thresholds and the resulting models have different ClinVar generalisation performance (AUC$_{0.1}$ = 0.67, AUC$_{0.01}$ = 0.73, AUC$_{0.001}$ = 0.65, AUC$_{0.0001}$ = 0.66). This shows that the chosen threshold does impact performance on a given task and suggests 0.01 is the best threshold for pathogenicity prediction, which aligns with the fact that the mean allele frequency for ClinVar benign variants is 0.112 and for pathogenic variants is 0.008 in the thousand genomes dataset [28].

Sequence UNET predictions would also be expected to relate to DMS results and confirmed neutral and deleterious *S. cerevisiae* variants. Comparing the distribution of Spearman's rank correlation values across DMS datasets (Fig. 3C) shows the model generalises, although it performs slightly less well than top predictors on this task. Performance is similar on datasets from different species, suggesting the model can be applied broadly (Additional file 1: Fig. S4). A similar result is found with a ROC analysis of S. *cerevisiae* variants (Fig. 3D), where Sequence UNET variations outperform FoldX but fall behind SIFT4G and ESM-1v. The model fine-tuned on ClinVar performs best in both cases, while the UNET top model performs relatively poorly, even falling behind simple CNN models in some cases. This further suggests that the UNET top model may be learning something more specific to ClinVar, which may be an artefact of the dataset but could also be a real biological feature of pathogenicity or human proteins. Together, this confirms that the models generalise well to other contexts, although their relative performance compared to other tools varies.

### High-throughput proteome scale predictions

Modern high-throughput experimental approaches can generate very large quantities of data, requiring efficient computational approaches to process. For example, a recent pan-proteome analysis by Muller et al. [18] collected protein abundance measurements from 103 species, detecting a total of 904,134 distinct proteins (Fig. 4A). Analysing this many proteins with the most commonly used predictors is very computationally intensive and would be prohibitively time consuming for many tools and research groups. For example, making predictions for 161,825 variants across just 30 proteins as part of a combined deep mutational scanning analysis [25] took SIFT4G 14.1 h and FoldX 64.5 days of total compute time. To exemplify the scalability of Sequence UNET, we made predictions for all 8.3 billion possible variants in this proteomics dataset, which took 1.5 h on a GPU using a batch size of 100 (6.8 h without batching) and 50.9 h using only CPU (Fig. 4B). The additional padding required to batch different length proteins was found to have a negligible impact on predictions for an analysis of this scale, although it does impact a small number of individual results (Additional file 1: Fig. S5). This also compares favourably with the ESM language model, even when using a single forward pass for all variants instead of independently masked passes for each variant as suggested [27]. ESM is both a much larger model and the attention mechanism scales quadratically with protein length, whereas the convolutional design of Sequence UNET scales linearly. This means Sequence UNET is significantly faster using CPU and for larger proteins on GPU. Small proteins are predicted at a similar rate on GPU, suggesting at this point other factors dominate. Sequence UNET also requires much less (V)RAM in all cases, making it significantly easier to deploy at scale and allowing batches of proteins to be processed simultaneously to increase efficiency. In contrast, even a batch size of 2 was prohibitive for ESM. These performance increases, combined with the Sequence UNET python package, makes large scale analyses more accessible, especially for those without high performance compute facilities.

We used this large dataset of variant effect predictions for almost 1 M proteins to compare protein abundance and predicted tolerance for sequence variation. Proteins that are expressed at higher abundances are generally expected to have more strongly

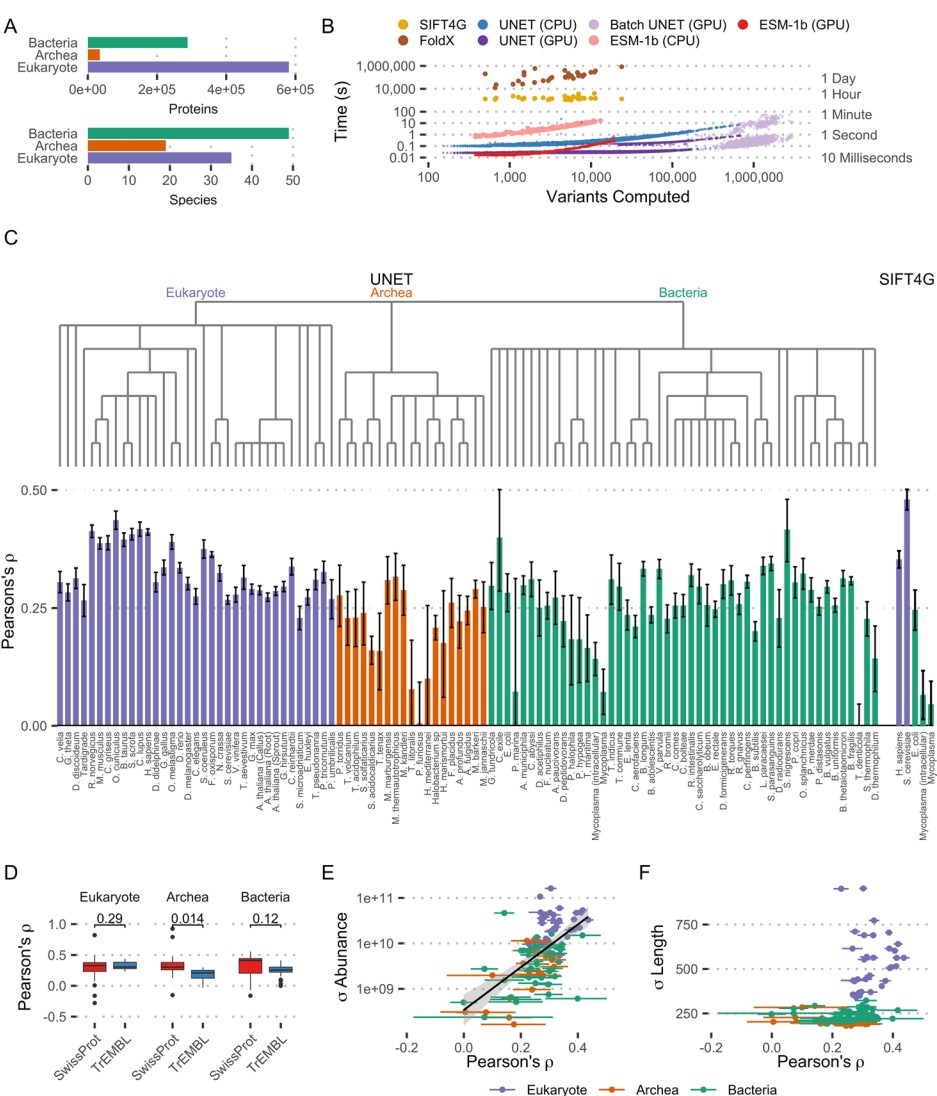

**Fig. 4** High-throughput proteome analysis. **A** Number of proteins and species in the Muller et al. proteomics dataset. **B** Computation speed comparison between SIFT4G, FoldX, single-pass ESM-1b and Sequence UNET on CPU and GPU. Sequence UNET was also tested running on GPU in batches of 100. The SIFT4G and FoldX computations were performed as part of an independent deep mutational scanning analysis [25], ESM-1b was run on ProteinNet proteins and Sequence UNET computations are across this proteomics dataset. **C** Pearson correlation coefficient between predicted conservation and protein abundance in each species. The error bounds of Pearson's $\rho$ are calculated with Fisher's $Z$ transform. Predicted conservation is summarised as the mean number of variants predicted to be deleterious across positions. Results are shown for Sequence UNET frequency predictions across all species and SIFT4G for Mycoplasma and species with data available in Mutfunc [29]. The species' phylogeny is also shown based on NCBI Taxonomy Common Tree. **D** Boxplot showing distribution of correlation coefficients for each domain, split between proteins in SwissProt and TrEMBL. The *p*-value comes from a two-sample unpaired *T*-test. **E** Relationship between Pearson correlation and standard deviation of raw protein abundance across species. **F** Relationship between Pearson correlation and standard deviation of protein length across species

constrained sequences than low abundance proteins [30]. This is thought to occur because highly expressed proteins need highly optimal sequences to avoid aggregation potentially driven by translation errors. However, most studies comparing abundance with sequence constraints have relied on a small number of species. In our

analysis, we found a significant correlation between protein abundance and predicted protein conservation in most of the 103 species in this dataset (Fig. 4C). Protein abundance is normalised against length and expressed as the $\log_2$ fold change compared to median abundance in that species. Conservation is summarised for a protein as the mean number of predicted deleterious variants across positions. A similar correlation level is observed for Sequence UNET predictions, including PSSM prediction and frequency classification, and SIFT4G scores in *H. sapiens*, *S. cerevisiae*, *E. coli* and Mycoplasma proteins. The similarity between correlations based on SIFT4G scores and our predictions validates the use of Sequence UNET for such applications.

The strength of conservation-abundance correlation varies a lot between species, including diminishing to nothing in a few cases. Eukaryotes tend to have the highest overall correlations (*t*-test vs bacteria: $p = 7.6 \times 10^{-6}$, vs Archaea: $p = 1 \times 10^{-5}$). However, looking more closely suggests this is partly caused by the fraction of TrEMBL proteins, which may include spurious open reading frames and, in our analysis, tend to have weaker correlations in Archaea (Fig. 4D). Variation in protein abundance (Fig. 4E) also impacts the abundance correlation, suggesting that part of the difference may come from reduced overall variation in protein forms. Eukaryotic proteins also tend to be more variable in length and have more consistently high correlation with abundance (Fig. 4F), again suggesting more variability impacts correlation. Species with no significant correlation tend to be unusual organisms, for example the intracellular parasite Mycoplasma or extremophile archaea such as *P. furiosus* or *T. litoralis*, which would be expected to have unusual properties based on their biology. They may also contain more proteins without similarities to those in our PDB based training data. There is also a stronger correlation in Mycoplasma when cultured intracellularly, suggesting it behaves more conventionally in that state and is more abnormal outside the cell. Finally, a similar correlation between abundance and conservation determined by SIFT4G scores shows a similar pattern in Mycoplasma, although with lower intracellular correlation, suggesting the low correlations have biological rather than technical causes. This simple analysis demonstrates the utility of performant, scalable predictors for large analyses and working with high-throughput experimental results.

## Discussion

Variant effect prediction (VEP) is a central part of many computational genetic analyses, allowing researchers to assess new genomes or patient sequences, prioritise variants for follow up experiments and identify important functional and structural features of proteins. This has led to a range of VEP tools, using empirical and machine learning methods to predict deleteriousness from sequence and structure. However, most tools are restrictive to run, particularly for large scale analyses involving hundreds of proteins. They can be computationally expensive, often having slow multiple sequence alignments or more recently machine learning model training as a limiting step, and are often awkward to install correctly. Precomputed results are available for human proteins and model organisms, for example in the Mutfunc [23, 29] or dbNSFP [31, 32] databases and the webpages of many tools. However, it is still difficult for many scientists to generate predictions for the proteins they need, especially when analysing a large number of proteins or organisms without pre-computed results. In an attempt to fill this niche,

we developed a versatile neural network VEP that is fast, lightweight and easy to use, able to rapidly make exhaustive predictions about many proteins on a regular laptop and scale to multi-proteome analyses on compute clusters. Similar lightweight approaches would also be beneficial for tools assessing many genetic backgrounds or combinations of mutations, where exhaustive pre-calculation is impossible.

Neural networks provide a powerful approach to fast, high-quality predictions, utilising large datasets and a long, computationally intensive training process to distil complex relationships into numerical weights matrices. This means predictions can be made efficiently as long as the network architecture uses optimised operations and does not require expensive external computations, for example multiple sequence alignment or structure relaxation. Neural network models have also shown very high performance in other sequence-based tasks [33–35] and variant effect prediction [12, 14], but there has not previously been a fully end-to-end neural network specialised VEP that operates directly on sequences. The Sequence UNET model architecture we arrived on takes inspiration from other CNN models, including from sequence, structure and image-based tasks [19], and combines them into a novel model. The U-shaped compression/expansion structure allows information to propagate across the protein, with the "receptive field" of neurons in the lower layers covering large regions of the original sequence in the same way they integrate information across images in the original UNET. This allows performant CNN operations to be used for a sequence-based problem while allowing filters to learn sequence patterns at different detail levels. Similarly, GraphC-NNs are performant and have been shown to perform on protein structure tasks [36–38], and their position invariance makes them a natural approach to including structural features. We experimented with various other methods of including structure, including torsion angles and calculated feature profiles, but found GraphCNNs gave best performance and efficiency.

The lack of labelled deleterious variant data at the scale required for deep learning led us to first capture general sequence and variant properties by training the model to predict variant frequencies, either as a PSSM predictor or a low frequency variant classifier, and then finetune for pathogenic variant classification using ClinVar data. This is similar to the protein language model paradigm, in which large models are first trained to predict amino acid sequences, capturing general properties, and the representation vectors they produce can be used as input into smaller downstream models. However, the size and design of the model makes it much more computationally efficient than most language models.

The base Sequence UNET model achieves state of the art performance at de novo PSSM prediction and frequency classification, both independently useful tasks, and the fine-tuned models reach top level performance at pathogenicity prediction, although the base model only generalises moderately well. There are also potential questions about performance on unstructured proteins, which are missing from ProteinNet, but this could be addressed by expanding the training data beyond structured proteins.

Variant frequency can be measured analytically and other models can predict pathogenicity to a broadly similar accuracy, meaning the major strength of our model is computational efficiency. This enables analyses to be completed more rapidly and on weaker hardware and opens up potential large-scale analyses that would not be possible

for more computationally demanding tools. For example, the multi-proteome analysis we performed here would have been extremely computationally expensive with many tools. Other possible applications include metagenome and microbiome analysis, where large numbers of new sequences are determined and need to be understood. Developing efficient models for structure, function or localisation prediction would also enhance work where a great many protein sequences are generated or need to be compared. Such methods can also be combined with slower, more accurate or analytical methods, identifying the most important targets for detailed, computation intensive analysis. This gets the benefits of speed and performance by enabling accurate results from the important areas of a wide search space.

There are various routes available that could improve our model in future, while maintaining computational efficiency. The role of protein structure is an obvious target for change, since it currently only adds a small performance boost despite structure being known to be critical for protein function. For example, a richer graph network section, graph attention mechanisms or pre-training the structural section to encode structural properties could all potentially improve performance. Removing the structure option altogether could also be beneficial because it would allow the model to be trained on a much larger sequence space, exposing it to more sequence variation and reducing the bias towards proteins with determined structures. The BFD used to train AlphaFold2 would be an appropriate dataset because it has already been processed into $\approx 66$ M unique sequence clusters with MSAs and minimal further processing would be required. It was derived from over 2.2B sequences taken from reference databases and collections and is over 600 times larger than ProteinNet. Large sequence datasets, and lots of training data in general, has been found to greatly improve performance in many other models [7, 17, 39]. Using sequences alone would make the model setup more similar to protein language models, which also train to predict amino acid propensity on large sequence databases. The UNET sequence CNN portion of the network could also be adjusted, either by tweaking the current connections and parameters or switching to an alternate sequence processing architecture. For instance, powerful attention and transformer architectures could be incorporated into the network or used as a basis for a new model that maintains computational efficiency as a goal, although it does come with an inherent computation cost compared to convolution. There are also alternate CNN architectures such as dilated convolutions [40], which can be combined with UNET designs [41]. These would also integrate information across the sequence but do not spread the receptive field as widely. However, they could also be added to the network and alternate approaches using them could also work for this problem.

## Conclusions

We have demonstrated a highly efficient, performant model for variant frequency and effect prediction, which enables larger scale analyses than have previously been possible with VEP packages, demonstrated by our multi-proteome conservation analysis. This could be beneficial for a range of biological problems where current models are slow enough to be prohibitive, for example metagenomics, microbiome research and analysing the large quantities of genomic sequences from the Darwin Tree of Life project [42]. More generally, developing computationally efficient deep learning models that

maintain high performance has great potential in other problems, speeding up predictions and providing approximate solutions where analytical approaches are prohibitive. This could enable new questions to be answered as well as making current analyses more economical, reducing compute time and consequently saving money and natural resources, including carbon [43].

## Methods

### UNET model

The Sequence UNET model is implemented in TensorFlow [44] (version 2.5 +) using the Keras framework. It takes a 3-dimensional tensor ($N \times M \times 20$) representing one-hot encoded protein sequences as input, with the final dimension being the amino acid, the middle protein position and the first outer dimension batches. As shown in Fig. 1, this is fed through $L$ compression CNN layers, where each layer contains two 1D CNN operations with $f \times 2^l$ filters on layer $l$ and width $k$ kernels. 1D max pooling operations compress the output of higher layers in half to input into the next layer. The input must be 0-padded in the second dimension so the length is divisible by $2L - 1$ times. The bottom layer contains a third CNN operation. $L$ corresponding expansion layers form the opposite side of the U structure, which each take input from a concatenation of the output from the corresponding compression layer and up-sampled output from the preceding expansion layer or the bottom layer. These also include 2 1D CNN operations. A final classification head processes the output of the final expansion layer, using an appropriate activation function for each task. Frequency classification uses a sigmoid function and frequency prediction the softmax function over the frequencies at each position. The final hyperparameters were $L = 6, f = 64, k = 9$, with ReLu activation functions for the CNN layers. In total this leads to 128 M parameters. The Swish activation function [45] was found to have slightly higher performance but technical details of TensorFlow 2.5 meant model gradients could not be loaded using Swish. In the future it could slightly improve performance but currently ReLU has been used to make distribution easier. Dropout at frequency 0.05 and batch normalisation were used in between each compression/expansion layer.

### Structural input

An optional simple GraphCNN [46] (Eq. (1)) is available to encode per position protein structure information, which is then concatenated with the one hot encoded sequence and fed into the main model. In this case, a weighted residue distance matrix (Eq. (2)) is required as an additional input. We tested various sizes, number of graph layers and residue distances weightings but found a single graph layer with 32 filters and ELu activation performed best.

$$H^{i+1} = \sigma \left( A H^i W^i \right) \tag{1}$$

$A$ is the $n \times n$ normalised edge matrix, which must be normalised to prevent the overall magnitude of features from changing and causing exploding or vanishing gradients. $H^i$ is an $n \times m$ matrix of the hidden values in layer $i$, for a graph with $n$ nodes with $m$ features. $W^i$ is an $m \times l$ matrix of learnt weights for layer $i$, where $l$ is the

number of features calculated for the next layer. This gives positional output that is a weighted sum of features from that and neighbouring amino acids, and operates independently of the input protein size.

The contact graph considers all residues within 10 Å to be in contact and uses a similarity metric (Eq. (3)) to weight closer residues more highly. A self-connection was used to pass information about the position itself and residues one or two positions away in sequence that were missing structural information were assumed to be 380 nm or 610 nm away, based on the average of non-masked positions. Other masked residues were assumed to not be in contact. This was one of the less explored areas of the network, so it is likely more complex structural features could improve performance.

$$a_{ij} = \frac{c_{ij}}{\sum_k c_{ik}} \tag{2}$$

$$c_{ij} = \begin{cases} \frac{1}{3+d_{ij}}, & d_{ij} \leq 10 \\ 0, & d_{ij} > 10 \end{cases} \tag{3}$$

where $d_{ij}$ is the distance between residues $i$ and $j$, $c_{ij}$ is the closeness between residues and $a_{ij}$ is the normalised residue contact matrix used in Eq. (1). Distances are measured in angstroms. The additional 3 Å means self-connections are weighted as roughly twice that of the nearest neighbour.

### ProteinNet dataset

We train the model using the ProteinNet dataset [21], which contains training, validation and test sets based on the CASP competition [22], with test sets containing the proteins predicted for each round of CASP and training sets containing all structures available in the PDB at the time of each competition. This does potentially introduce a slight bias into the test sets since only proteins with newly discovered crystal structures are included in CASP competitions, meaning they tend to be more unusual and less well studied proteins or those that are difficult to crystallise. However, the test sets still contain diverse proteins from a range of organisms, so any reduction in generalisation is likely to be small and predictions would generally be expected to be worse on these unusual proteins that cover novel sequence space. We use the CASP12 95% thinned training set as a balance between including a large variety of slightly different sequences and moderating the dataset size for rapid training. CASP is designed to assess template-based structure prediction methods [47] so a minority of proteins in the test set (38 of 146 in CASP12) have some sequence similarity to proteins in the PDB, which could potentially artificially inflate test performance by a small amount. However, the size of the training dataset means the model is unable to memorise variant consequences and so having a minority of somewhat related sequences in the test set is very unlikely to significantly influence results. Together, this makes ProteinNet a convenient and appropriate dataset for predicting deleteriousness from sequence and structure, despite that not being its original purpose, and using it greatly sped up

model development. We developed the ProteinNetPy package [48, 49] to help parse, process and manipulate data from ProteinNet. This makes it easier to use variants of the model that require structural data as well as helping use ProteinNet for other work.

### Training and hyperparameter optimisation

The model was trained and assessed using 32 GB Nvidia Tesla V100 GPU nodes on a high-performance computer cluster. ProteinNetPy was used to generate the TensorFlow datasets for training and validation, using custom map functions depending on the model configuration being trained. A progressive hyperparameter optimisation strategy was used, testing a range of values for each parameter and using the best results as the default value of that parameter for future tests. The tested parameters were number of UNET layers, number of filters, kernel width, presence of structure features, optimisation algorithm, deleteriousness threshold, activation function and regularisation regime (see Additional file 1: Fig. S1). These were tested for both PSSM prediction and frequency classification modes of the model, with the same resulting parameters. When not testing optimisers or their settings, training was performed using the Adam optimiser [50] with a learning rate of 0.01 and early stopping based on validation accuracy with a memory of 20 epochs. Increasing model size tends to increase performance, so training speed and resource consumption is balanced directly against performance. We settled on 64 first layer filters, doubling in each layer; 9 wide kernels; and 6 layers to comfortably train on a single Nvidia Tesla V100 GPU; the size could be increased using more resources to incrementally improve performance. Increasing the number or size of GraphCNN layers did not improve performance, suggesting more advanced techniques [7, 36] might be required to fully harness structure. A single 32 filter GraphCNN layer was used to encode structural features.

We trained two versions of the model, optimising hyperparameters independently for each case. We first trained to predict the frequency of each possible mutation in a protein and therefore its PSSM. The Kullback–Leibler divergence between the vector of predicted and true frequencies of each position was used as the loss function, and a softmax activation function was used in the final layer. We next trained the model as a frequency classifier, predicting whether each possible variant occurs below a frequency threshold. This used a binary cross-entropy loss function and a sigmoid activation function in the final layer. We tested a range of frequency thresholds (Fig S3), settling on $f = 0.01$. Further details of the performance and characteristics of these networks can be found in Additional file 1: Supplementary Information. The training code [51] contains the exact specification of the training procedures used. Model predictions were generated for fasta and ProteinNet data using the sequence_unet python package, from the same GitHub repository.

### Baseline CNNs

Three sizes of baseline convolutional model were also trained for comparison: a single layer network with 32 7 width filters, an equivalent double layer network and a larger network with 64 7 width filters in the second layer. Training was performed with the ProteinNet CASP12 dataset, using the 95% data split.

### SPBuild predictions

We generated SPBuild predictions for all proteins in the ProteinNet CASP 12 test set using the latest version of SPBuild (2020–01-07). Sequence UNET naturally outputs raw variant frequencies, whereas PSSMs are generally reported as log scores normalised against WT amino acid frequencies (Eq. (4)), as SPBuild reports. Therefore, the frequencies output by Sequence UNET, the baseline CNN and ESM-1b models were transformed into the standard PSSM format to compare them to BLOSUM62 and SPBuild.

$$\log_2 \frac{f_{\text{pred}} + 10^{-5}}{f_{\text{AA}}} \tag{4}$$

where $f_{\text{pred}}$ is the predicted frequency and $f_{\text{AA}}$ the average frequency of that amino acid, based on Swiss-Prot summary statistics [52, 53].

### ESM-1b and ESM-1v predictions and derived models

We used the pre-trained ESM-1b 650 M and ESM-1v 650 M models to generate logits and representations of protein positions for comparison to Sequence UNET. For more direct comparisons, we also trained single layer PSSM prediction (softmax activation) and frequency classifiers (sigmoid activation) on top of the ESM-1b representations using the ProteinNet CASP12 95% thinning data.

### SIFT4G results

SIFT4G was run on all the proteins in the ProteinNet CASP12 testing and validation sets, using a modified version of SIFT4G that outputs 5 decimal places instead of 2.

### Model generalisation

The 24/04/2021 ClinVar dataset was used for training and testing model generalisation. ClinVar classifications were simplified to pathogenic or benign, each category including variants designated as likely or definitely benign/pathogenic. Fine-tuned models were trained to predict ClinVar pathogenicity, either by replacing the classification head layer (top model) or refining its weights (finetuning). In both cases, only the classification head and the preceding layer are trainable with the rest of the model frozen, which means only a small number of weights are trained. The models take wild-type sequence or sequence and structure as input and the final fully connected layer outputs a matrix of pathogenicity scores for all possible variants using sigmoid activation. During training, variants which are not in the ClinVar training data are masked and do not contribute to training. Fine-tuned models in both styles were trained based on both PSSM prediction and frequency classification models with 3 and 1 width convolution kernels. They were trained using binary cross entropy loss and the Adam optimiser for a maximum of 50 epochs with 10 epoch memory early stopping. In practice validation performance decreased after 10 to 15 epochs in each case, so weights from this stage were selected by the early stopping procedure. Three wide kernel models based on the classification model performed best and were used.

Additional generalisation comparison datasets were downloaded or computed using the relevant tool. A combined set of deep mutational scanning results was sourced from our previous work [25]. DMS ER scores less than -0.5 were classified as deleterious. A

gold standard set of *S. cerevisiae* variants was downloaded from work by Jelier et al. [26]. SIFT4G and FoldX predictions for ClinVar variants were retrieved from Mutfunc [29], ESM-1v scores were calculated directly, EVE scores were downloaded from its website and all other tool results were retrieved from dbNSFP [31, 32].

### Proteomic analysis

The sequences of proteins identified in the Muller et al. pan-proteome analysis were downloaded from UniProt. The sequence_unet predict_from_fasta command was then used to generate Sequence UNET predictions using only sequence input for the PSSM prediction and frequency classification models. Prediction results for each protein were summarised as the mean number of predicted deleterious variants across positions. Protein abundance was normalised against length and expressed as the $\log_2$ fold change compared to median abundance in that species. The phylogeny of these organisms was downloaded from NCBI taxonomy [54, 55]. SIFT4G scores for comparison were downloaded from Mutfunc [29] for *H. sapiens*, *S. cerevisiae* and *E. coli* and calculated using Uniref90 and default settings for Mycoplasma.

### Supplementary Information

---

**Additional file 1: Supplementary Information.** This file contains a longer description of the python packages, supplementary analyses, and all the supplementary figures and relevant discussion. It includes details of the performance of the base PSSM prediction and frequency classifier model and discussions of dataset biases and the effect of variable padding. It also contains some more description of the python packages.

**Additional file 2.** Review History.

---

#### Acknowledgements
Not applicable.

#### Review history
The review history is available as Additional file 2.

#### Peer review information

#### Authors' contributions
AD designed and coded the model, performed analysis and wrote the manuscript. PB and MA supervised, helped design experiments and reviewed the manuscript. All authors read and approved the final manuscript.

#### Funding
 AD is funded by Wellcome Grant 220540/Z/20/A, 'Wellcome Sanger Institute Quinquennial Review 2021–2026'. P.B. is supported by the Helmut Horten Stiftung and the ETH Zurich Foundation.

#### Availability of data and materials
All data and code used in this study is available in public repositories. A python package and weights for easily using and implementing the model as well as development code are available on GitHub [51] under the Apache Licence v2.0. The code at time of release is archived on Zenodo [56]. The weights are also available in BioStudies (S-BSST732) [57]. This code fully defines the network and training procedures. We also developed a python package for loading and manipulating ProteinNet data (ProteinNetPy) [48], also under the Apache Licence v2.0. The code at time of release is archived on Zenodo [49]. ProteinNet [21] and ClinVar [58] data is available online and the deep mutational scan [25] and yeast pathogenicity [26] data is available from their respective publications.

### Declarations

#### Ethics approval and consent to participate
Not applicable.

**Consent for publication**
Not applicable.

**Competing interests**
The authors declare that they have no competing interests.

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

## 