## [**Additional file 2.** Review History. · Genome Biology]

Review History

First round of review

Reviewer 1

Were you able to assess all statistics in the manuscript, including the appropriateness of statistical tests used? Yes

Were you able to directly test the methods? No

Comments to author:

The main selling-point of this method is the low run-time and lower computational costs compared to other modern methods such as EVE/DeepSequence/ESM.

I like the U-shaped architecture that allows integration of long and short-range features

Figure 1 - It seems that panels B, C and D are a bit redundant. B summarises how 1D convolution works and includes a formula but doesn't explain the components. C just re-hashes the model architecture shown in A. D doesn't seem to fit with anything else and once again has a formula with no components explained.

- Panel E: Is it really necessary to have the initialisation time and prediction time as different bars? While I understand that the initialisation only happens once per session, these results are for one session (SARS-COV2 Spike). Just stack them and have a 85.42 second bar for UNET.

"The majority of VEPs fall within this range, with most requiring a computation intensive step such as structural sampling" -- I don't disagree with this statement, but ESM-1v should be considerably quicker than the 22 minutes taken by SIFT4G. I don't see any reason not to include it in this comparison since it is discussed at length later.

Pre-computed results (dbNSFP) are discussed briefly in the discussion section but doesn't really answer the question of why I would want a really fast predictor if I can get pre-calculated ClinPred (which came top on the ClinVar data for pathogenic predictions in humans).

The authors suggest that UNET might be most useful on less-studied organisms. However, the ClinVar analysis only applies to humans, and the DMS correlation analysis is not split by organism. I don't believe there is enough evidence to say UNET would be valuable in non-human scenarios (especially since it underperformed SIFT against yeast variants).

The top model architecture and training is not described in any detail.

Figures 2 and 3: would love an idea of the sample sizes for the ROC and PR AUC values either in the text or figure legend.

- Training the top model: were steps taken to prevent type 2 data circularity? I.E. ensuring that each protein has all its variants EITHER in the testing OR training set (not split). What about closely related proteins?

- There is no commentary about the performance of ClinPred, REVEL and the other VEPs that out-performed UNET on the ClinVar data in Figure 3a. Could data circularity be influencing the predictions of these predictors?

- Coloring in Figure 3 is confusing. UNET takes pink -> blue -> purple -> red (PreGraph UNET* is almost grey and blends in with the other grey bars). Yellow appears to be methods previously mentioned in the paper ('notable methods'). Colouring by methodology (or even run time!) would be less confusing.

Figure 4B - Red (UNET GPU) and dark orange (ESM-1b CPU) are near-indistinguishable.

Panel 4F is not referenced, it also makes it seem like there is little relation between variance in length and the correlation, contrary to the text.

Reviewer 2

Were you able to assess all statistics in the manuscript, including the appropriateness of statistical tests used? Yes

Were you able to directly test the methods? No

Comments to author:

- Are the methods appropriate to the aims of the study, are they well described, and are necessary controls included?

Yes, the methods seem appropriate for the study. The authors use an innovative UNET architecture for coding variant effect prediction (VEP) on a variety of tasks. They compare their method against a wide range of VEP methods which provides a better understanding of the strengths and weaknesses of their tools compared to other tools.

- Are the conclusions adequately supported by the data shown?

Yes, the conclusions seem to be supported by the data they show. They provide comparisons with a nice representative sample of VEP methods ranging from evolutionary to structure-based as well as recently developed protein language models. They compare their method to these other methods on a variety of benchmarks. They explain where their model does well and when other methods seem to outperform it.

- Are sufficient details provided to allow replication and comparison with related analyses that may have been performed?

They provide a great deal of detail on their experimental setup and benchmarks but I found a few aspects of their modeling confusing, namely:

What is the "representative protein" being used for these models?

How are the variant frequencies being calculated for each "representative protein"?

Are all of the fine-tuning tasks being trained in the same way as the PSSM and frequency training?

My intuition is that they are using wild-type protein as the input and then you are predicting variant frequency/PSSM at each position. However, I read the following sentence and then became confused

"We trained ClinVar classification top models and fine-tuned models with and without structural features on top of the Sequence UNET frequency classification model, using a random 95%/0.5%/4.5% training/validation/testing split across all pathogenic and neutral variants in the ClinVar dataset that occur in proteins in ProteinNet CASP12 training data."

This sentence made me think that you are inputting mutagenized proteins into your model. It may be clear if you are an expert in the field, but I think it would be beneficial to the broader audience to add further detail on how the inputs for these models are being generated and if you are using the same type of input for each of the downstream tasks. Perhaps a simple flow chart showing how the input protein sequence is generated would be helpful.

- Is the method likely to be of broad utility? Is any software component easy to install and use? Please indicate briefly the novel features and/or advantages of the method, and/or please reference the relevant publications and which methods, if any, it should be compared with.

Yes, this paper contains a novel and efficient UNET architecture which seems to be able to quickly predict VEP for large numbers of variants and proteins. They also demonstrate its utility by calculating VEP for the very large "Tree of Life" dataset. They also discuss ways for potentially improving their model on tasks where their model isn't a top performer. Therefore, the novel, efficient architecture and potential for improvement could make this framework very useful to a wide variety of groups interested in VEP for proteins. The authors also provide a straightforward python API for users to download the model weights and score variants from ProteinNet. It would be good, for future use, for the API to be able to score proteins in formats that are not in an ML-friendly format such as data within ProteinNet.

- Is the paper of broad interest to others in the field, or of outstanding interest to a broad audience of biologists?

I believe this paper would be of interest to both machine learning researchers that are interested in exploring applications of ML to proteins and geneticists that are interested in calculating VEPs for their protein of interest. This is why I believe it would be helpful for the authors to clarify the points I raise about model training.

Clarity/Presentation:

The major issue I have with the presentation is a misunderstanding of aspects of your model training that I have raised previously. I think it would be helpful to have a simple diagram explaining how you get your "representative protein" that is used as input into your model. If I understand your modeling correctly, I think it would be helpful to emphasize that for a given protein you have aggregated all variant information for that protein and then you are using that as a single training example. For example, with the ClinVar data, are you taking a particular protein and then classifying every amino acid in every position as pathogenic/benign? I was confused about that when reading the text.

I also found the usage of "top models" confusing when discussing UNET models where the feature extraction part is fixed and the classification head is replaced. There are other times when "top model" refers to the top scoring model among the various models used within your benchmarks. I would suggest renaming them in order to convey that you are simply fixing the feature extractor part.

Major Concerns:

I found it interesting that the baseline CNN model did so well on the PSSM and frequency prediction tasks. I am wondering if a slightly larger CNN model with dilated convolutions could also have achieved the same receptive field width as the U-Net model and with same performance as your U-Net model. Could you comment on why the U-Net model would be a preferable approach than something like a dilated CNN model?

Minor Concerns:

Define "PTMs"

Fix the order of the AUCs for different MAF thresholds

MAF for "pathogenic variants is 0.008": this is surely very dependent on ascertainment.

In the "Generalising Sequence UNET" section the authors state

"The top model performs slightly better than the fine-tuning approach on variants with structure but worse on those without it, independently of whether the network utilised the information. This is perhaps because the freshly initialised final weights allow the network to learn new relationships specific to structured pathogenicity whereas fine-tuning maintains more general relationships"

I find this statement counter-intuitive because I would have thought fixing the feature extraction portion would constrain the new relationships that can be learned. Can you provide me some intuition for why that might not be the case?

Conclusion:

I would recommend the paper for acceptance in Genome Biology provided rectifications to the concerns relayed in the clarity/presentation, minor concerns, and major concerns sections. The authors do a nice job benchmarking their methods against a variety of other methods and a variety of other tasks. It is also particularly refreshing that the authors have developed a resource efficient computational tool which can be widely used by academic groups.

Please find below our point-by-point response to the comments of the reviewers. Our responses are in blue and we have also marked in blue in the main text the significant text changes to make it easier for reviewers to see where changes were made.

Review Comments

Reviewer #1

The main selling-point of this method is the low run-time and lower computational costs compared to other modern methods such as EVE/DeepSequence/ESM.

I like the U-shaped architecture that allows integration of long and short-range features.

We appreciate the positive remarks from the reviewer. We agree with the assessment that the strongest benefit of this approach is the fast run-time and ease-of-use of the method and tool. We also think that the application constitutes an interesting novel biological analysis performed when comparing the protein abundance levels with the predicted impact of mutations across the tree of life.

Figure 1 - It seems that panels B, C and D are a bit redundant. B summarises how 1D convolution works and includes a formula but doesn't explain the components. C just re-hashes the model architecture shown in A. D doesn't seem to fit with anything else and once again has a formula with no components explained.

We agree that 1 B, C and D are somewhat redundant and have revised the figure accordingly.

- Panel E: Is it really necessary to have the initialisation time and prediction time as different bars? While I understand that the initialisation only happens once per session, these results are for one session (SARS-COV2 Spike). Just stack them and have a 85.42 second bar for UNET.

We think it is quite relevant to make the difference between initialisation and prediction time very obvious since one of the strengths of the approach here is to be able to run large scale predictions where the initialization time becomes less important. We have left panel E as it is but we are willing to change it if the reviewer feels strongly about this.

"The majority of VEPs fall within this range, with most requiring a computation intensive step such as structural sampling" -- I don't disagree with this statement, but ESM-1v should be considerably quicker than the 22 minutes taken by SIFT4G. I don't see any reason not to include it in this comparison since it is discussed at length later.

In the particular case of this example, the Spike protein (1273 residues) is too large to run using the ESM-1v model, which is limited to proteins of length 1022 residues or less.. Since the speed benchmarking was done more extensively in Figure 4B, we could remove this comparison in panel E but we wanted to highlight at the start the speed benefits of using our model.

Pre-computed results (dbNSFP) are discussed briefly in the discussion section but doesn't really answer the question of why I would want a really fast predictor if I can get pre-calculated ClinPred (which came top on the ClinVar data for pathogenic predictions in humans).

As we illustrate in the application part of the manuscript, our method allows for proteome-wide predictions across many species. There are many applications, such as the one we tried, where this is required. We have revised the discussion to further emphasise that the main utility of these approaches is for non-human proteomes where pre-computed variant effects are not easily accessible.

The authors suggest that UNET might be most useful on less-studied organisms. However, the ClinVar analysis only applies to humans, and the DMS correlation analysis is not split by organism. I don't believe there is enough evidence to say UNET would be valuable in non-human scenarios (especially since it underperformed SIFT against yeast variants).

We do not expect the model to perform differently on different species as it was not trained on data specific to any one species. Much of the DMS data was itself collected in yeast, even though we are assessing protein coding genes from other species. Nevertheless, we followed the reviewer's suggestion and we split the DMS analysis by species of origin for the proteins tested. We show the results for UNET below, where performance doesn't vary significantly across species. We have also added this figure to the supplementary.

The top model architecture and training is not described in any detail.
We have added details about this in the results and methods sections.

Figures 2 and 3: would love an idea of the sample sizes for the ROC and PR AUC values either in the text or figure legend.

In figure 2 all analysis is done on the 40 proteins in the ProteinNet test set, with 12094 total amino acids, giving 241880 total variants. In figure 3b there are ~500-700 ClinVar variants, 3c >9000 up to >100000 DMA variants depending on tool and in 3d there are ~6000+ Yeast variants of which ~2000 have structural data for FoldX. We have added these to the figures/legends.

- Training the top model: were steps taken to prevent type 2 data circularity? I.E. ensuring that each protein has all its variants EITHER in the testing OR training set (not split). What about closely related proteins?

Splits were randomised by variant without accounting for source protein. However, the make up of the ClinVar dataset means only a small proportion of variants in the test set come from proteins also represented in the training data. Similarly, fine-tuning was run only for a small number of epochs (<20) with early stopping, limiting the ability to overfit on the source protein. At the point that early stopping

occurs training and validation loss do not diverge. We have added a discussion of this issue to the ClinVar section of the supplementary.

- There is no commentary about the performance of ClinPred, REVEL and the other VEPs that out-performed UNET on the ClinVar data in Figure 3a. Could data circularity be influencing the predictions of these predictors?

Many of these are ensemble predictors (now shown in the figure with colouring based on another suggestion) and as such have often been exposed to the same proteins previously in some manner, making it hard to know whether their results are due to overfitting or not. We have added a discussion of this issue to the results section.

- Colouring in Figure 3 is confusing. UNET takes pink -> blue -> purple -> red (PreGraph UNET* is almost grey and blends in with the other grey bars). Yellow appears to be methods previously mentioned in the paper ('notable methods'). Colouring by methodology (or even run time!) would be less confusing.

We agree this was confusing and have updated the colours to follow the methodology group.

Figure 4B - Red (UNET GPU) and dark orange (ESM-1b CPU) are near-indistinguishable.

We have revised the colour scheme to make these have greater contrast.

Panel 4F is not referenced, it also makes it seem like there is little relation between variance in length and the correlation, contrary to the text.

We have added a specific reference to 4F and clarified our interpretation in the text.

Reviewer #2

- Are the methods appropriate to the aims of the study, are they well described, and are necessary controls included?

Yes, the methods seem appropriate for the study. The authors use an innovative UNET architecture for coding variant effect prediction (VEP) on a variety of tasks. They compare their method against a wide range of VEP methods which provides a better understanding of the strengths and weaknesses of their tools compared to other tools.

We thank the reviewer for the positive remarks about our work.

- Are the conclusions adequately supported by the data shown?

Yes, the conclusions seem to be supported by the data they show. They provide comparisons with a nice representative sample of VEP methods ranging from evolutionary to structure-based as well as recently developed protein language

models. They compare their method to these other methods on a variety of benchmarks. They explain where their model does well and when other methods seem to outperform it.

We thank the reviewer for the positive remarks about our work.

- Are sufficient details provided to allow replication and comparison with related analyses that may have been performed?

They provide a great deal of detail on their experimental setup and benchmarks but I found a few aspects of their modeling confusing, namely:

What is the "representative protein" being used for these models?

How are the variant frequencies being calculated for each "representative protein"?

Are all of the fine-tuning tasks being trained in the same way as the PSSM and frequency training?

My intuition is that they are using wild-type protein as the input and then you are predicting variant frequency/PSSM at each position. However, I read the following sentence and then became confused

"We trained ClinVar classification top models and fine-tuned models with and without structural features on top of the Sequence UNET frequency classification model, using a random 95%/0.5%/4.5% training/validation/testing split across all pathogenic and neutral variants in the ClinVar dataset that occur in proteins in ProteinNet CASP12 training data."

This sentence made me think that you are inputting mutagenized proteins into your model. It may be clear if you are an expert in the field, but I think it would be beneficial to the broader audience to add further detail on how the inputs for these models are being generated and if you are using the same type of input for each of the downstream tasks. Perhaps a simple flow chart showing how the input protein sequence is generated would be helpful.

Thanks for bringing this point of confusion to our attention. We have revised the results section to make it more accessible to a wider audience. In all cases the model input is the same (wild-type sequence and optionally structural data), which we have now emphasised at several points in the main text and in the expanded methods section. The fine-tuned and top models train a small model on top of the original Sequence UNET output that is specialised for pathogenicity, a distinction that we have also clarified in the text.

- Is the method likely to be of broad utility? Is any software component easy to install and use? Please indicate briefly the novel features and/or advantages of the method, and/or please reference the relevant publications and which methods, if any, it should be compared with.

Yes, this paper contains a novel and efficient UNET architecture which seems to be able to quickly predict VEP for large numbers of variants and proteins. They also demonstrate its utility by calculating VEP for the very large "Tree of Life" dataset. They also discuss ways for potentially improving their model on tasks where their model isn't a top performer. Therefore, the novel, efficient architecture and potential for improvement could make this framework very useful to a wide variety of groups interested in VEP for proteins. The authors also provide a straightforward python API for users to download the model weights and score variants from ProteinNet. It would be good, for future use, for the API to be able to score proteins in formats that are not in an ML-friendly format such as data within ProteinNet.

The python package already supports a range of sequence formats, accepting BioPython sequences and raw strings. Compatibility with BioPython makes it easy to import sequences from most common formats. Structural data does need to be in ProteinNet format, but this covers most PDB entries and we provide the ProteinNetPy package that allows you to load and filter these files. The package also includes a script that scores Fasta (and ProteinNet) files without any code, which hopefully makes it easy for users.

We thank the reviewer for pointing out that this wasn't made clear and have outlined the capabilities of the packages in the method section.

- Is the paper of broad interest to others in the field, or of outstanding interest to a broad audience of biologists?

I believe this paper would be of interest to both machine learning researchers that are interested in exploring applications of ML to proteins and geneticists that are interested in calculating VEPs for their protein of interest. This is why I believe it would be helpful for the authors to clarify the points I raise about model training.

We appreciate the positive feedback and have tried to improve the clarity of the manuscript as indicated by the reviewer.

Clarity/Presentation:

The major issue I have with the presentation is a misunderstanding of aspects of your model training that I have raised previously. I think it would be helpful to have a simple diagram explaining how you get your "representative protein" that is used as input into your model. If I understand your modeling correctly, I think it would be helpful to emphasize that for a given protein you have aggregated all variant information for that protein and then you are using that as a single training example. For example, with the ClinVar data, are you taking a particular protein and then classifying every amino acid in every position as pathogenic/benign? I was confused about that when reading the text.

The reviewers interpretation is correct and we have now clarified at several points in the text that the model takes an input sequence and outputs a matrix of predictions scoring each possible substitution amino acid for each position.

I also found the usage of "top models" confusing when discussing UNET models where the feature extraction part is fixed and the classification head is replaced. There are other times when "top model" refers to the top scoring model among the various models used within your benchmarks. I would suggest renaming them in order to convey that you are simply fixing the feature extractor part.

Top model is the term generally used when doing transfer learning of this type (i.e. training a new model on top of the output of another), so we believe changing it to a new term wouldn't improve clarity in this context. We agree however that there could be some confusion here and have altered the text so that "top model" is only used in the manner we describe above. Furthermore we now specify which model each top model derives from (e.g. UNET top model or ESM1b top model) where appropriate. We hope this has improved clarity without changing accepted technical terms or terminology deeply embedded in the code.

Major Concerns:

I found it interesting that the baseline CNN model did so well on the PSSM and frequency prediction tasks. I am wondering if a slightly larger CNN model with dilated convolutions could also have achieved the same receptive field width as the U-Net model and with same performance as your U-Net model. Could you comment on why the U-Net model would be a preferable approach than something like a dilated CNN model?

We have added some discussion on other CNN architectures to the discussion. Effectively UNET is a larger CNN model with additional architectural features that capture long range effects, similar to dilated convolutions. Indeed it would be possible to use both (e.g. <https://arxiv.org/ftp/arxiv/papers/2104/2104.03130.pdf>). Our expectation is that with further optimisation, it is possible to obtain good performance with a model based on dilations and one incorporating both may in fact do best. However, designing and optimising other variations of a CNN design would be a significant project in itself.

Minor Concerns:

Define "PTMs"

We have changed to say post-translational modifications since we never actually use the PTM acronym.

Fix the order of the AUCs for different MAF thresholds

We have changed AUCs to be in decreasing MAF order. Previously they were in decreasing AUC order, but we find the new version more intuitive.

MAF for "pathogenic variants is 0.008": this is surely very dependent on ascertainment.

That is true. In our case, we are reporting the observed frequencies from the thousand genomes dataset. We have reworded the text to make this clear.

In the "Generalising Sequence UNET" section the authors state

"The top model performs slightly better than the fine-tuning approach on variants with structure but worse on those without it, independently of whether the network utilised the information. This is perhaps because the freshly initialised final weights allow the network to learn new relationships specific to structured pathogenicity whereas fine-tuning maintains more general relationships"

I find this statement counter-intuitive because I would have thought fixing the feature extraction portion would constrain the new relationships that can be learned. Can you provide me some intuition for why that might not be the case?

In both cases the feature extraction part of the network is frozen, with only the last two layers of weights being trainable. This does constrain the new relationships that can be learnt, which is deliberate because the purpose of transfer learning is to apply the general features learnt from the larger ProteinNet dataset to the smaller ClinVar data. The difference between the top model and the fine-tuned model is that the weights in the top model are randomly initialised whereas the fine-tuned model starts from the final weights of the main Sequence UNET. This means it starts from a configuration that captures general features and fine-tunes it for pathogenicity. In contrast the top model starting weights don't capture anything and so are freer to be trained towards specific minima that relate to structured essential human proteins only. We have tried to reword this sentence to make it more informative.

We hope these responses help answer reviewers questions and we thank them for their useful feedback.

Second round of review

Reviewer 1: I am happy with the authors' response to my comments, and think the paper is now suitable for publication.

Reviewer 2

The authors have satisfactorily addressed the issues I raised in my initial review. I would recommend this manuscript for acceptance into the journal with some minor improvements and clarifications. I will trust the authors to address these points without me needing to see the manuscript again.

1. The caption for Figure 3B is confusing, the authors state “Variants of Sequence UNET are coloured in purple, blue and red, single layer CNN models in green and several notable models in yellow.” But the models in blue and red are not variants of Sequence UNET. Consider modifying the text so it is not confusing.

2. In the section “Generalising Sequence UNET” the authors state “This is perhaps because the freshly initialised final weights allow the network to learn new relationships specific to the pathogenicity of structured human essential proteins in ClinVar whereas fine-tuning starts from weights which capture general relationships and refines these rather than reaching highly specific minima.” I find this statement confusing because my impression is the “top” model is using the UNET model as a feature extractor which is then refined on the task at hand whereas the fine-tuning is jointly adjusting the feature extraction and task-specific parts of the model. Could it not just be that fine-tuning has not converged to the correct minima whereas the “top” model was able to converge since the feature extraction part was fixed? Could this phenomenon be a function of the training data size? It would be worth addressing this if this were the case.

3. In the conclusion as well as at different points of the manuscript you make the point “There are also potential questions about performance on unstructured proteins, which are missing from ProteinNet, but this could be addressed by expanding the training data beyond structured proteins.” Could you address the types of datasets that may exist which would mitigate this bias within the ProteinNet dataset? Are there publicly available that already exists for VEP prediction or would one have to generate these datasets? I think it would be of interest to the community to understand that better.

Overall, I think this is an excellent paper that both raises and addresses some very important points for variant effect prediction of proteins. As deep learning models get larger and more computationally expensive, it is refreshing for researchers to think about efficiency and wider dissemination to researchers who may not have the computational resources on par with large companies.